# Improved Biological Phosphorus Removal under Low Solid Retention Time Regime in Full-Scale Sequencing Batch Reactor

Ghazal Srivastava [1,*], Aparna Kapoor [2] and Absar Ahmad Kazmi [1]

1    Environmental Engineering Group, CED, IIT Roorkee, Roorkee 247667, Uttarakhand, India
2    SFC, C. Tech. SBR Technologies Limited, Mumbai 400705, Maharashtra, India
*    Correspondence: ghazalsrivastava1247@gmail.com

**Abstract:** Enhanced biological phosphorus removal (EBPR) is an obscure but economical and helpful technology for removing phosphorus biologically from wastewater. A 3-MLD capacity pre-anoxic selector-attached sequencing batch reactor (SBR) treated municipal wastewater from the residents of IIT Roorkee. The treatment in the plant satisfied the effluent discharge standards in all respects except phosphorus, observed during an intensive two-year study. An elaborated 80-day study was performed to enhance and improve the plant's performance in terms of phosphorus removal specifically, with run 1: solid retention times (SRT) reduced from 56 to 20 days (t = 35 d), run 2: lowering the diffuser's running time from 15 min to 10 min in anoxic cum anaerobic selector chambers (dissolved oxygen (DO) concentration reduced to <0.15 mg/L) along with reducing SRT to 15 days (t = 25 d), and run 3:intensive reduction in SRT to $\leq$10 days (t = 20 d). During run 3, the increment in the enhanced biological phosphorus removal (EBPR) efficiency was three times that of the initial run ($\eta_{max}$~65%) with a readily biodegradable chemical oxygen demand to total phosphorus ratio (rbCOD/TP) of 7.8. The 16SrRNA sequencing revealed the microbial community structure before and after the changes in SRT and EBPR efficiencies, to correlate the biochemical processes and functional organisms.

**Keywords:** biological phosphorus removal; sequencing batch reactor; simultaneous nitrification-denitrification; solids retention times

## 1. Introduction

The enhanced biological phosphorus removal (EBPR) process seems convoluted due to peculiar microbial sensitiveness but has been a useful technology since the 1950s for removing phosphorus biologically from wastewater. It needs two phases: anaerobic and aerobic zones for the continuous function of phosphorus release and uptake, respectively, in systems. Despite the complexity of EBPR mechanisms, appropriately proposed, designed, and operated treatment plants can easily accomplish phosphorus (P) removal on the condition that EBPR-available organic substrates, such as short-chain volatile fatty acids (VFAs) and anaerobic–aerobic (respectively) reactor arrangements are offered.

Decreasing the solid retention time (SRT) of the EBPR process can intensify organic carbon diversion to the side stream for energy recovery and phosphate release. Determining the least (i.e., minimum) SRT for polyphosphate accumulating organisms (PAOs) or denitrifying PAOs in simultaneous phosphorus and nitrogen removal accompanied by carbon diversion for energy recovery from EBPR systems is crucial [1]. Moreover, microbial community analysis using multiple molecular and Raman techniques suggested that the relative abundances and diversity of PAOs and glycogen-accumulating organisms (GAOs) in EBPR systems are significantly affected by varying the SRT [2]. The resultant EBPR process stability and performance can be potentially controlled and optimized by running the system's SRT, and shorter SRTs (<10 days) are preferable [3,4].

A study on the activated sludge-treatment process suggested that activated sludge with an SRT of 16 days acquired better nitrite nitrogen utilization, organic matter degrada-

tion, and phosphorus removal than sludge with an SRT of 6, 10, and 22 (days); polyphosphate concentration in anaerobic activated sludge per volatile suspended solids (VSS) decreased to 23.1 mg/g VSS; and poly-β-hydroxy butyrate (PHB) concentration in anoxic activated sludge per VSS was reduced to 16.3 mg/g VSS; therefore, a spectacular denitrifying phosphate removal occurred [5,6]. However, another study investigated the effects of SRT on sludge characteristics and operational performance in an EBPR reactor in which the results showed that the reactor operated at SRT of 8.3 days could achieve a phosphate removal efficiency of >90% and a sludge volume index (SVI) of <100 mL g$^{-1}$ [7]. In comparison, increasing SRT to 16.6 days led to a decrease in phosphate removal <85% and an increase in SVI value (160 mL g$^{-1}$), implying a performance degradation and worse settling ability of the sludge [7]. Though the relationship between SRT and P removal has not yet been satisfactorily examined and an evaluation of the consequence of SRT on microbial populations and associated performance on full-scale EBPR systems is inadequate, it has been observed in a few studies that lower SRTs benefit EBPR systems [3,8,9]. Therefore, it is necessary to examine the optimum SRT for maximizing the efficiencies of biological nutrient removal in full-scale SBR-based sewage treatment plants in India.

In full-scale wastewater-treatment plants, the dominating EBPR community accompanying nitrogen removal are *Candidatus Accumulibacter*, *Tetrasphaera*, *Nitrosomonas*, and denitrifiers such as *Zooglea*, *Paracoccus*, *Pseudomonas* with proteobacteria, bacteroidota, and actinobacteria phylum [10,11]. It has been established that maintaining conditions that support the proliferation of PAOs over GAOs is essential for the stability of EBPR systems [2,12,13].

Assessment of full-scale EBPR performance records by [2,13] specified that process stability was positively correlated with the incoming wastewater's C/P ratio. They proposed that an optimized process can be maintained, even if the extreme amount of existing carbon can encourage the propagation of GAOs, given that the operational conditions are managed to favor the augmentation of PAOs over GAOs. However, the very low readily biodegradable chemical oxygen demand (rbCOD)/TP was found to be insufficient for PAO growth. Values ranging from 10 to 20 are observed to be adequate for the prevalence of PAOs over GAOs [13,14]. Several other factors, in addition to the carbon-to-P ratio, can influence the competition between PAOs and GAOs, and they include SRT, substrate form, hydraulic retention time (HRT), temperature, pH, dissolved oxygen (DO), and feeding strategy [3,15–17]. Among the known factors that could influence EBPR population dynamics, SRT, DO, and feeding patterns are more governable to implement in a full-scale setting for actual practice. It was suggested that the increase in SRT could lead to the reduction in biomass yield and excess sludge released, therefore reducing the P removed [18]; hence, lower SRTs are observed to be favorable for the EBPR mechanism.

SRT is primarily considered an important factor affecting the performance of the nutrient removal (nitrogen and phosphorus both) and sludge characteristics, and the production of secondary pollutants such as nitrous oxide (N$_2$O) in biological nutrient removal (BNR) processes [19]. For nitrogen removal, some studies also suggest that the nitrifying bacteria communities in activated sludge may change significantly with variations in the SRT and substrate concentration [20,21]. According to [22], at an SRT less than or equal to 20 days, the effluent ammonia concentration was lower than the effluent nitrite concentration; however, the effluent nitrite concentration became equal to or less than the effluent ammonia concentration at 40 days' SRT. Quantitative polymerase chain reaction (QPCR) assays indicated that increasing SRT significantly increased the nitrite-oxidizing bacteria/ammonia-oxidizing bacteria (NOB/AOB) ratio. In the study by [19], the removal of chemical oxygen demand or total phosphorus was similar under SRTs of 5–40 days; however, SRT mainly affected the nitrogen removal, and the optimal SRT for BNR was 20 days.

However, the studies by [23] demonstrated that the decrease in SRT from 37.2 to 27.8, 19.0, and 10.0 days had no significant influence on the development of simultaneous nitrification and denitrification (SND). Their study constituted the system of a continu-

ous flow-activated sludge (AS) process and they observed its behavior under different solid retention times (SRT), food-to-mass (F/M) ratios, and dissolved oxygen (DO) concentrations. Their records showed that even the DO $0.5 \pm 0.2$ mgO$_2$/L did not affect the growth of heterotrophic bacteria (2.2–3.1 d$^{-1}$ at 20 °C) and autotrophic nitrifying bacteria (0.16–1.94 d$^{-1}$ at 20 °C), resulting in similarities with conventional AS processes in their study. Hence, both SND- and EBPR-derived technologies require an optimum SRT regime to work efficiently.

Therefore, the objectives of this study were to improve the P-removal conditions biologically from a full-scale SBR plant accomplishing high COD, BOD, TSS, and nitrogen removal (via SND), simultaneously, from wastewater based on the stringent National Green Tribunal (NGT) and Central Public Health and Environmental Engineering Organization (CPHEEO) guidelines. Varying, i.e., reducing SRTs (by removing dead and inactive excess sludge having low PO$_4$-P release and PO$_4$-P uptake capabilities) and enhancing anaerobic conditions of pre-anoxic/anaerobic selector zones could transform a system into a better one for EBPR. Identifying an optimum SRT for good EBPR along with high nitrogen removal is tricky, but manageable in full-scale existing plants. Three different runs were shown in the study including four purposes:

- To study the influence of lowering SRT (up to 10 days) on EBPR in SBR;
- To study the influence of lowering SRT (up to 10 days) on SND in SBR;
- Using molecular biological techniques for the identification of microbial communities in SBR (before and after lowering the SRT);
- To study the effect of dewatered sludge characteristics on reducing SRT in SBR.

Selector-attached SBR systems are excellent and flexible for improvising and advancing the plant for BNR [24]. This system was comprehendible and practical for all pre-anoxic selector-attached SBR-based full-scale treatment plants in India working at low rbCOD/P ratios.

## 2. Material and Methods

### 2.1. SBR technology and Overall Process

The SBR technology and process are demonstrated in Figure 1.

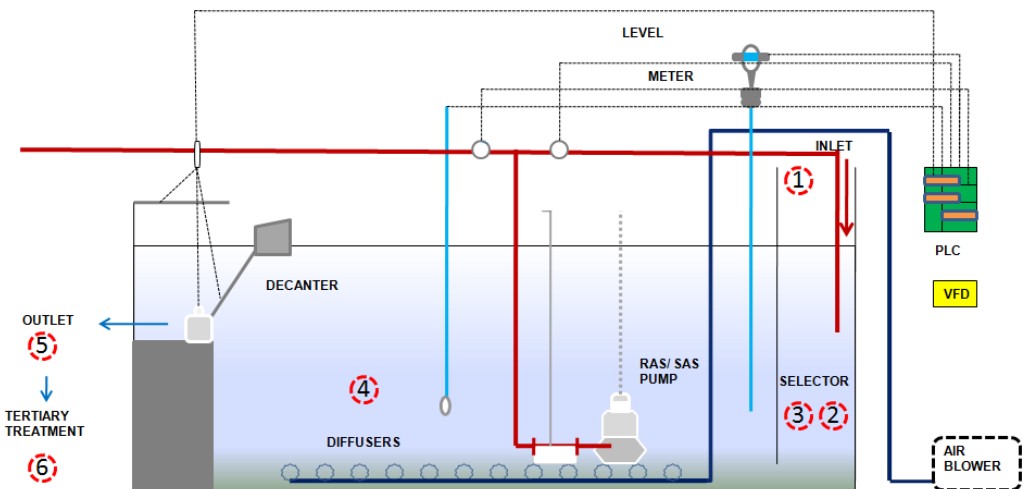

**Figure 1.** Pre-anoxic selector-attached SBR process of 3 MLD SBR plant. The encircled points in figure depict—1: Inlet (raw sewage), 2: Anoxic selector zone (1st)-where returned activated sludge (RAS) and inlet (raw sewage) combine directly, 3: Anoxic cum anaerobic selector zones (2nd and 3rd compartments), 4: SBR Aeration zone, 5: Biologically treated outlet, and 6: Finally treated outlet after tertiary treatment (disc filtration and UV disinfection).

The 3-MLD SBR has been set up in a neighborhood close to a residential area in front of the IIT, Roorkee campus, Uttarakhand (India). The essential features of this STP are

the deodorization system's additional odor control for the sump well, pre-treatment units, and advanced tertiary treatment facility (fiber disc filtration and ultraviolet radiation and chlorination for disinfection). The onsite monitoring of various parameters was performed in the bio-selectors and aeration tanks of the 3-MLD SBR plant and the design configurations of the plant were stated in our published paper by [25].

Onsite monitoring of DO, pH, ORP, and $SV_{30}$ is executed regularly in the bio-selectors and aeration tanks of the 3 MLD SBR Plant. To determine the DO, temperature, and pH in the aeration tanks and selectors, a portable DO meter (Hach 110Q multimeter, Hach, Loveland, CO, USA) and a pH meter (HQ11d pH Meter, Hach) were used [26]. ORP was measured by the convenient ORP meter (HQ11d ORP Meter, Hach). Complete performance evaluation of the plants for COD, sCOD, $BOD_5$, $sBOD_5$, TSS, VSS, $NH_4$-N, $NO_3$-N, TN, $PO_4$-P, TP, total coliforms, fecal coliforms, and sludge operational parameters were performed according to standard methods [27]. The rbCOD was determined using the modified flocculation filtration method prescribed by [28]. $SV_{30}$ was measured using the measuring cylinder and timer [25]. Grab samples of 0.5L were used for analyzing the parameters mentioned above, according to standard methods [27]. Regression analysis of all the variables was performed using the Microsoft Excel spreadsheet application using F-test and ANOVA.

### 2.2. Wastewater Characteristics

The design quality of raw sewage for the SBR plant is shown in [25]. The ratio between VSS to TSS was found around $0.57 \pm 0.05$. The designed flow rate and HRT were 3.2 MLD and ~18.2 h. The raw wastewater pH was $7.1 \pm 0.2$, and finally, treated effluent after disinfection was $7.4 \pm 0.1$. Regular sampling and analysis were conducted continuously for three years in this plant, and a run-wise study was performed for three months. The average daily flow rate was $2372 \pm 168$ m$^3$/d. Detailed wastewater characterization was also performed for COD, BOD, TSS, TN, and TP before and after SRT. Table 1 depicts several runs to increase the EBPR of the plant. After reducing the SRT, the parameters changed, and the results are described in Tables 2 and 3, and Figure S1 (Supplementary Materials).

**Table 1.** Operational parameters including EBPR and SND in different runs.

| S. No. | Runs/Phases | Phase Description | Duration | References |
|:---:|:---:|:---:|:---:|:---:|
| 1. | Initial phase | SRT > 50 days | 2–3 years | [24,25] |
| 2. | Run-I | Modifications include lowering the plant's solid retention times (SRT) from 56 to 20 days | 35 days | Present study |
| 3. | Run-II | Lowering the diffuser's running time from 15 min to 10 min in anoxic cum anaerobic selector chambers (DO concentrations reduced to <0.15 mg/L) along with reducing SRT from 20 days to 15 days | 25 days | Present study |
| 4. | Run-III | Combined effects by lowering the diffuser's running time from 15 min to 10 min in anoxic cum anaerobic selector chambers (DO concentrations reduced to <0.15 mg/L) with an intensive reduction in SRT from 15 days to ≤10 days | 20 days | Present study |

**Table 2.** Performance assessment and temporal variations of influent and effluent of 3 MLD SBR in different runs and at variable SRTs.

| Parameters | At High SRT | | Run 1 | | Run 2 | | Run 3 | |
|---|---|---|---|---|---|---|---|---|
| | Influent | Effluent | Influent | Effluent | Influent | Effluent | Influent | Effluent |
| SRT (d) | 56.8 ± 6.4 | | 25.8 ± 9.3 | | 19.7 ± 2.4 | | 11.4 ± 2.6 | |
| pH | 7.2 ± 0.1 | 7.3 ± 0.1 | 7.0 ± 0.3 | 7.4 ± 0.1 | 7.1 ± 0.4 | 7.4 ± 0.2 | 7.0 ± 0.2 | 7.4 ± 0.2 |
| Alkalinity (mg/L) | 320 ± 40 | 220 ± 40 | 340 ± 60 | 240 ± 50 | 320 ± 40 | 210 ± 60 | 340 ± 50 | 230 ± 80 |
| COD (mg/L) | 401 ± 51 | 17 ± 3.8 | 387 ± 45 | 20 ± 5.6 | 377 ± 52 | 18.9 ± 5.1 | 356 ± 60 | 21.5 ± 4.0 |
| sCOD (mg/L) | 121 ± 31 | 8.8 ± 4.0 | 111 ± 25 | 9.0 ± 3.5 | 96 ± 40 | 9.2 ± 3.1 | 124 ± 32 | 9.8 ± 1.3 |
| BOD (mg/L) | 168 ± 22 | 7.9 ± 2.1 | 156 ± 31 | 7.1 ± 3.0 | 187 ± 25 | 6.5 ± 1.8 | 175 ± 30 | 6.6 ± 1.8 |
| sBOD (mg/L) | 51 ± 23 | 3.1 ± 1.2 | 48 ± 31 | 3.2 ± 2.0 | 49 ± 11 | 2.8 ± 1.1 | 56 ± 14 | 3.1 ± 1.0 |
| $NH_4$-N (mg/L) | 22 ± 3.0 | 0.3 ± 0.3 | 24 ± 3.5 | 0.4 ± 0.3 | 23 ± 2.1 | 0.5 ± 0.4 | 28 ± 4.5 | 0.7 ± 0.3 |
| $NO_3$-N (mg/L) | 0.2 ± 0.1 | 5.7 ± 0.2 | 0.4 ± 0.2 | 5.5 ± 1.0 | 0.52 ± 0.5 | 6.6 ± 1.3 | 0.7 ± 0.4 | 7.1 ± 1.3 |
| TKN (mg/L) | 31 ± 3.3 | 4.0 ± 1.2 | 33 ± 4.0 | 3.4 ± 2.8 | 32.9 ± 1.3 | 3.3 ± 2.3 | 37.6 ± 3.5 | 2.7 ± 2.2 |
| TN (mg/L) | 31.2 ± 3.3 | 9.7 ± 1.5 | 33.4 ± 4.1 | 8.9 ± 3.0 | 33.4 ± 3.3 | 9.9 ± 2.1 | 38.3 ± 4.0 | 9.8 ± 3.2 |
| $PO_4$-P (mg/L) | 2.8 ± 0.2 | 2.5 ± 0.1 | 2.8 ± 0.3 | 2.0 ± 0.3 | 2.7 ± 0.8 | 1.3 ± 0.4 | 3.1 ± 0.5 | 1.1 ± 0.3 |
| TP (mg/L) | 4.5 ± 0.6 | 3.1 ± 0.1 | 4.4 ± 0.4 | 2.7 ± 0.4 | 4.6 ± 0.7 | 2.0 ± 0.4 | 4.8 ± 0.6 | 1.7 ± 0.4 |
| TSS (mg/L) | 261 ± 31 | 9.9 ± 4.1 | 252 ± 34 | 9.4 ± 3.5 | 251 ± 31 | 9.2 ± 6.0 | 237 ± 33 | 9.1 ± 5.2 |
| VSS (mg/L) | 149 ± 13 | 5.2 ± 3.1 | 141 ± 32 | 4.9 ± 3.2 | 145 ± 14 | 5.4 ± 1.3 | 134 ± 31 | 5.2 ± 3.1 |
| Total Coliforms (MPN/100 mL) | 240,000 ± 5200 | 1712 ± 150 | 210,000 ± 9100 | 1500 ± 240 | 340,000 ± 12,900 | 1209 ± 50 | 480,000 ± 9500 | 990 ± 60 |
| Fecal Coliforms (MPN/100 mL) | 1300 ± 78 | 79 ± 21 | 1100 ± 54 | 55 ± 22 | 2100 ± 43 | 47 ± 11 | 1800 ± 60 | 49 ± 13 |

**Table 3.** Operational sludge characteristics in the main aeration zone and the last anoxic–anaerobic selector compartment of SBR.

| Sludge Parameters | At High SRT | | RUN 1 | | RUN 2 | | RUN 3 | |
|---|---|---|---|---|---|---|---|---|
| | Aeration | Selector | Aeration | Selector | Aeration | Selector | Aeration | Selector |
| pH | 7.3 ± 0.1 | 7.4 ± 0.2 | 7.4 ± 0.1 | 7.6 ± 0.1 | 7.2 ± 0.2 | 7.5 ± 0.1 | 7.4 ± 0.2 | 7.8 ± 0.2 |
| ORP (mV) | 95.3 ± 7.9 | −33.1 ± 15.0 | 84.6 ± 12 | −59.1 ± 18.4 | 95 ± 6.2 | −93 ± 14.7 | 121.2 ± 27 | −129.4 ± 22.1 |
| DO (mg/L) | 1.5 ± 0.5 | 0.4 ± 0.2 | 1.6 ± 0.5 | 0.3 ± 0.1 | 2.1 ± 0.3 | 0.2 ± 0.02 | 2.5 ± 0.5 | 0.1 ± 0.02 |
| Temperature (°C) | 29.5 ± 0.5 | 29.4 ± 0.4 | 29.1 ± 0.7 | 29.2 ± 0.4 | 29.0 ± 0.6 | 28.8 ± 0.5 | 28.5 ± 1.0 | 28.6 ± 1.2 |
| MLSS (mg/L) | 5761 ± 1787 | - | 3389 ± 756 | - | 2766 ± 319 | - | 1682 ± 352 | - |
| MLVSS/MLSS | 0.50 ± 0.02 | - | 0.52 ± 0.02 | - | 0.59 ± 0.04 | - | 0.63 ± 0.03 | - |
| SVI (mL/g) | 55.3 ± 15.7 | - | 65.0 ± 8.4 | - | 71.1 ± 6.5 | - | 74.5 ± 7.8 | - |
| HRT (h) | 20.5 ± 0.9 | - | 18.5 ± 1.4 | - | 18.2 ± 1.2 | - | 17.7 ± 1.1 | - |

Wastewater characterization as COD and TP Fractions.

(1) COD Fractions in wastewater (as mg/L)

COD fractionation involves the identification of inert and biodegradable COD together with readily biodegradable and slowly biodegradable fractions. COD fractions were observed in four forms: readily biodegradable COD (rbCOD), slowly biodegradable COD (sbCOD), non-biodegradable soluble COD (nbsCOD), and non-biodegradable particulate COD (nbpCOD). Experimental methods developed or selected for the assessment of all the COD fractions should be compatible with the mathematical models defining biological treatment and yield consistent and reliable values [29]. The inert particulate COD goes to the sludge produced, and the inert soluble fraction of COD goes to the treated effluent. The rbCOD content takes part in the metabolism and hydrolysis inside the bacterial cell, whereas sbCOD has the potential to metabolize but at slower rates. The rbCOD content

helps the microorganisms with a quick metabolism that gets fermented as volatile fatty acids (VFAs) in the anaerobic zone. The fractions of sbCOD and rbCOD are also involved in PHB production during the anaerobic phase. The rbCOD is a critical parameter for biological process performance, specifically undergoing SND and EBPR in treatment plants [24,30].

However, a proper interpretation of rbCOD is conducted using a floc-filtration method used in the study [28]. Other fractions of COD were determined according to [31]. The rbCOD was ~11.7%, sbCOD was ~71.2%, nbsCOD was ~2.6%, and nbpCOD was ~14.5% of total COD in the wastewater. The soluble portion of COD was ~30.7% of total COD and soluble BOD was ~26.9% of total BOD (Figure 2). The COD, BOD, TSS, and TP fractions at high SRT is present in Figure S2 (Supplementary Materials).

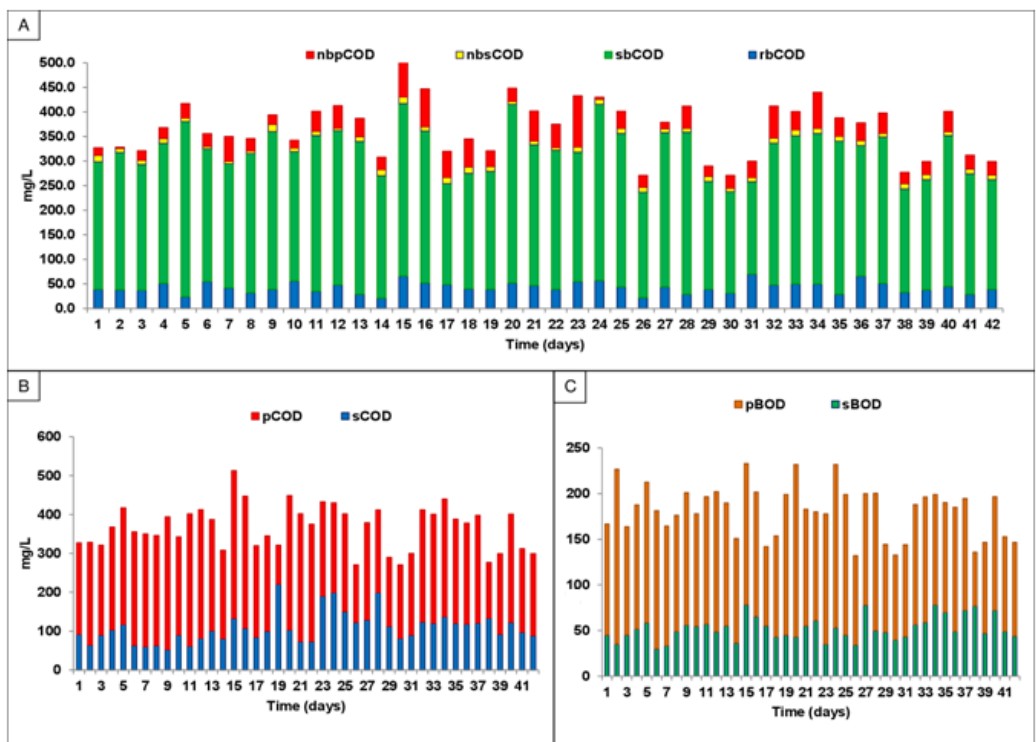

**Figure 2.** COD and BOD fractions in wastewater (SRT 10–30 days) in the plant (**A**) all COD Fractions broadly classified as 'biodegradable' and 'non-biodegradable' (**B**) 'soluble' and 'particulate' forms of COD (**C**) all BOD fractions- soluble (sBOD) and particulate (pBOD) ('y axis' depicts 'mg/L').

In Figure 2, COD and BOD fractions are shown in detail. The fraction of rbCOD in total COD is observed as responsible for nitrate reduction and PHB synthesis during SND and EBPR pathways, respectively. The overall fraction of rbCOD was 11.7 ± 3.1% in the SBR. Figure 2B illustrates the soluble and particulate portions of COD as sCOD (soluble COD) and pCOD (particulate COD), respectively. The soluble COD was observed as only 31.3 ± 14.4% of total COD. Similarly, BOD fractions are also characterized as soluble (sBOD) and particulate (pBOD). The soluble BOD fraction was 26.5 ± 9.1% of total BOD, representing that suspended biodegradable organic matter was higher in the influent wastewater and is metabolized slowly during the bacterial activities.

(2) TP Fractions in wastewater (as mg/L)

The Indian standards for phosphorus limits are stringent for biological wastewater-treatment units (≤1 mg/L). Most sewage-treatment plants (STPs) are observed to be effective in enhanced biological phosphorus removal (EBPR) and biological nitrogen removal processes simultaneously. There is a valuable role to play for anaerobic cum anoxic selectors and advanced, cost-effective SBR technologies in simultaneous biological phosphorus and nitrogen removal in the coming years in India. Therefore, an in-depth study of different phosphorus fractions and their removal mechanisms is needed to justify and

upgrade the design processes. The different phosphorus fractions are SRP (soluble reactive phosphorus), PRP (particulate reactive phosphorus), SOP (soluble organic phosphorus), POP (particulate organic phosphorus), SAHP (soluble acid-hydrolysable phosphorus), and PAHP (particulate acid-hydrolysable phosphorus) (Figure 3). The different fractions of TP were determined using standard methods [27,32], and according to [33].

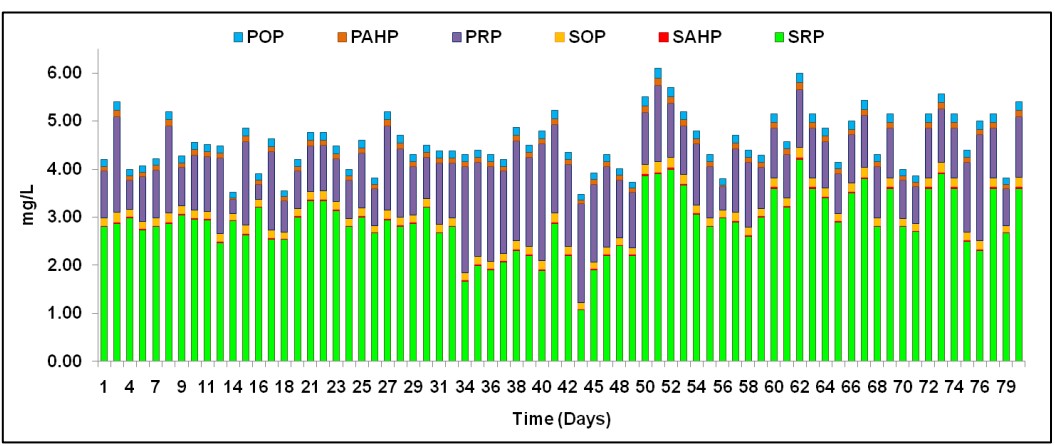

**Figure 3.** TP fractions in wastewater coming into the 3 MLD SBR plant.

The soluble reactive phosphorus (soluble orthophosphates) constitutes the major portion in TP, 63.1% of total phosphorus, and significantly participates in EBPR phenomena. The particulate reactive phosphorus (PRP) was around $1.2 \pm 0.5$ mg/L (26.9%). The SOP and SAHP were only 3.6% and 0.7% of total phosphorus in the influent, showing that dissolved organic phosphorus and dissolved polyphosphates remained low in sewage, and particulate organic and polyphosphates, contributing to 3.4% and 2.3%, respectively, mostly adhered to the settled portion of influent wastewater.

### 2.3. Microscopic Investigations for Intracellular Polymers Development, Protozoa, and Microbial Identification

There are specific intracellular polymers, i.e., PHBs and polyphosphates, which are produced in the cells of polyphosphate-accumulating organisms (PAOs) and denitrifying polyphosphate-accumulating organisms (DPAOs) during the EBPR mechanism. During EBPR and SND, the PHBs are formed in the anaerobic phase or pre-anoxic zones of the SBR cycle and get degraded when the organism needs that stored carbon substrate for metabolism in the corresponding aerobic phase. Polyphosphates are formed in the aerobic phase by PAOs or DPAOs. The PHBs are detected through light microscopic analysis using Sudan Black B staining by light microscope (B-383PHi, Optica, Milan, Italy) and STEM microscopy using an Apreo field emission scanning electron microscope with a low vacuum. Polyphosphate globules were detected using DAPI staining using Fluorescence Microscope B-383, Optica, Italy, and Neisser staining by light microscope, respectively [18,34]. The bright-field images for intracellular polymer identification were captured at 100X magnification and protozoa and other microbe identification was performed through optical microscopy at phase contrast 10X, 20X, 40X, and 100X magnifications, and scanning electron microscopy (SEM) analysis with a Carl Zeiss Gemini field emission scanning electron microscope.

### 2.4. 16SrRNA Sequencing and DNA Extraction

The DNA extraction was performed to extract the DNA from the samples using a DNA extraction soil kit (Xploregen discoveries, Bengaluru, India) and then the readings were observed in a nanotube. After the quantification, the results for PCR were analyzed, and then 16SrRNA Illumina sequencing was performed for 16S primers at a particular cycle time described in the Supplementary Materials. The protocol used for metagenomics was as follows: (1) deoxyribonucleic acid (DNA) extraction and the quality check, (2) PCR product



purification and polymerase chain reaction (PCR) amplification with V3-V4 primers, and (3) sequencing using the Illumina MiSeq Platform. Data analyses were exhibited and a brief explanation is given in the Supplementary Materials. The statistical analyses were performed using Microbiome analyst software and Venn diagram was plotted using Venny 2.1 software.

### 2.5. Dewatered Sludge Parameter Analyses

The dewatered sludge parameters were checked and analyzed by standard methods [27]. Some of the analyses such as SOUR, density, and calorific value were determined using [31]. The pathogens salmonella, fecal coliforms, and helminth eggs were identified using [27,35].

### 2.6. Energy Consumption

Energy consumption was recorded daily from the PLC screen of the SBR plant. The results were compared at different DO values with SRT changes. The SRT, SAS pump running time, blower's running time in selector zones, and MLSS concentration in aeration tanks was observed as critical factors to affect the energy consumption of the plant which ultimately affects the electrical energy audits and operation and maintenance costs. Lowering the SRT to 10–12 days renders the system economic and maintains it under control. It has been stated in the literature that high SRTs (>30 days) lead to an increased oxygen requirement for the plant and increased energy usage or high energy-consumption rates (KWh/d) of the plant, whereas very low SRTs (<5 days) effect elevated ammonia and elevated BOD in the treated effluent [36]. Therefore, establishing the optimum SRT is the ultimate requirement for satisfying all criteria, including excellent SND, conventional nitrification and denitrification, EBPR, final settling, COD removal, reduced energy consumption, and low biological sludge production.

## 3. Results and Discussion

### 3.1. Average Temporal Variations at Different SRTs in the SBR Plant

The overall performance assessment of the SBR plant shows that the COD, BOD, TSS, $NH_4$-N, TN, and fecal coliforms were removed effectively at different SRTs. However, the removal of total phosphorus was increased (from $30.6 \pm 7.3\%$ to $61.2 \pm 7.6\%$) by lowering the SRTs (from $46.8 \pm 6.4$ days to $11.4 \pm 2.6$ days). The pathogens in the effluent, i.e., fecal coliforms, were also reduced on decreasing SRT. The reason could be that the plant conditions are optimized at lower SRTs (10–12 days).

### 3.2. Operational Sludge Parameters

The operational sludge characteristics were observed in different runs (Table 3). The MLSS in different runs were controlled by wasting excess sludge from the SBR tanks to optimize the SRT (Figure S3, Supplementary Materials). As the MLSS was maintained between 1000–2000 mg/L concerning constant sludge settling behavior, the SVIs were observed to be increased from ~55 mL/g to ~75 mL/g at lower SRTs, i.e., <10 days. Varying SRT, i.e., reducing up to 10–12 days, led to a decrease in BOD, TSS, and phosphorus in the effluent; however, nitrates were observed to be slightly increased (from 5.7 mg/L to 7.1 mg/L) in the effluent because of incomplete denitrification at low SRTs. At higher SRTs, the $PO_4$-P uptake rates were reduced.

As the sludge remains in the aeration tanks for a long time, endogenous decay rates were higher at elevated SRTs, and the sludge loses its capacity to further uptake orthophosphate in the form of polyphosphates. Hence, excess P uptake is not possible in that biomass, and it is required to throw the dead and neck-filled phosphorus sludge and let the SBR form new biomass. Newly produced biomass can have the capability to accumulate the excess P and perform EBPR by maintaining optimal, low-SRT conditions [36].

### 3.3. Biological Nutrient (N and P) Removal at Variable SRT

High SRT is required for carbonaceous oxidation, ammonia oxidation, and sludge reduction, while low SRT is preferred for enhanced phosphorus assimilative uptake [9,37–39].

For nitrogen removal, high SRTs are generally preferred in initial phases. At high SRT (>50 days), the ammonia-N removal was 99.0 ± 1.6%; at run 1, 2, and 3 (26 days, 20 days, and 11 days average SRT, respectively) the removal efficiencies were 98.8%, 98.3%, and 97.4%, respectively, and $NO_3$-N in the effluent increased, i.e., 5.5 mg/L, 6.6 mg/L, 7.1 mg/L, respectively, while in the anoxic selectors, some denitrification occurred; 2.1 mg/L, 1.3 mg/L, and 1.3 mg/L, respectively (Figure 4). The denitrification rates achieved 37% by lowering the SRT in the selector zones and decreasing the diffuser running rates from 1/4th of an hour to 1/6th of an hour during a cycle, i.e., maintaining the DO values at <0.14 mg/L. The SND rates slightly decreased to 78%, 73%, and 77% in runs 1, 2, and 3, respectively. For $PO_4$-P removal, by decreasing SRT the removal rates increased 28.0%, 51.0%, and 59.1% at run 1, 2, and 3 (26 days, 20 days, and 11 days average SRT, respectively); similarly the TP removal efficiencies also increased 38.9%, 57.1%, and 61.2%, respectively. The EBPR% improved by 14.5%, 17.8%, and 25.7% in runs 1, 2, and 3, respectively.

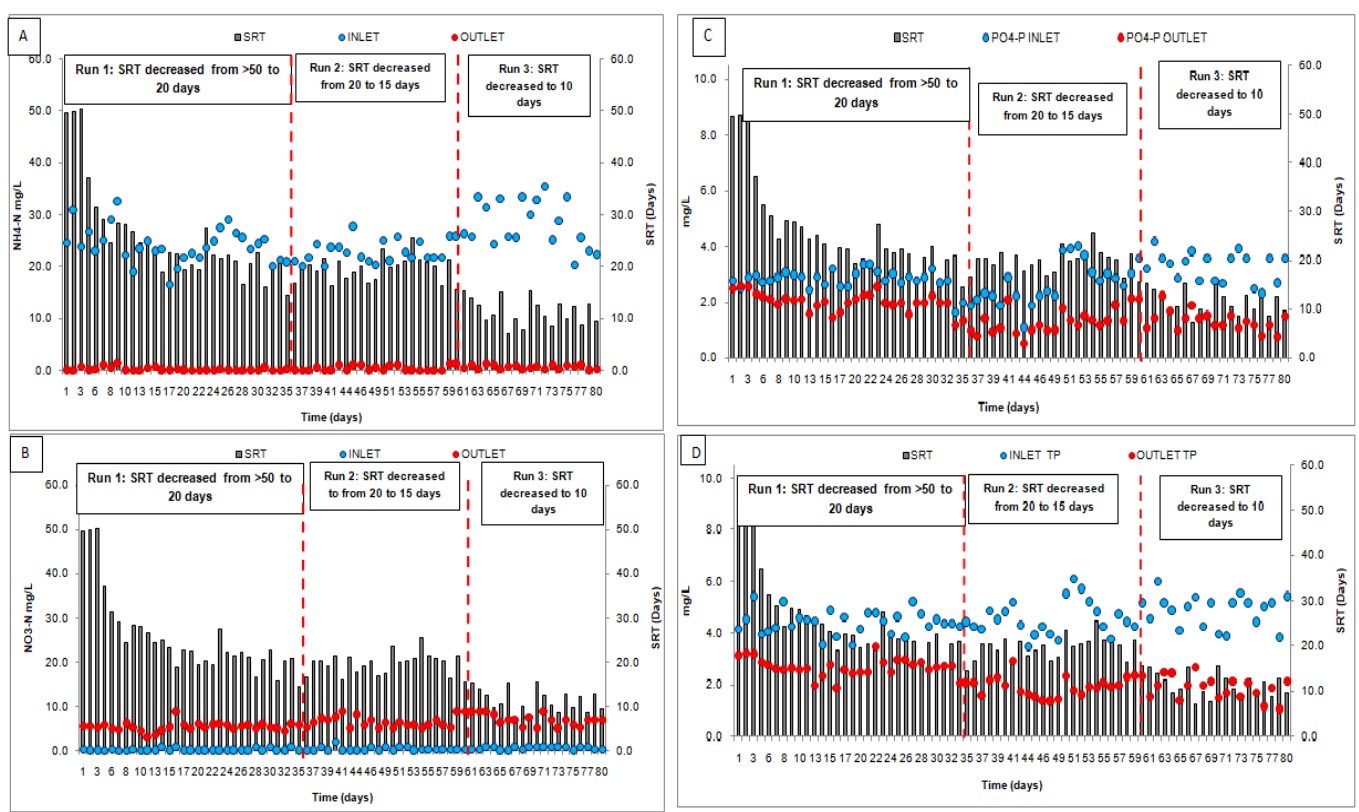

**Figure 4.** Run-wise profiles of (**A**) $NH_4$-N, (**B**) $NO_3$-N, (**C**) $PO_4$-P, and (**D**) TP in 3 MLD SBR plant.

Figure 5A shows the relationship between $PO_4$-P and TP removal with SRT. The statistical analyses (Table 4) show that dissolved phosphorus removal with SRT has a linearly decreasing relationship which is statistically significant ($R^2$ ~0.5, $p < 0.001$) and TP removal is also statistically dependent on SRT with decreasing function equation ($R^2 > 0.34$ and $p < 0.001$). The statistical analyses are exhibited in Table 4. Additionally, Figure 5B depicts that minimizing SRT earlier than 10 days did not greatly affect the SND%; the relationship was constant, but the reduction in SRT slightly affected nitrate reduction (nitrate levels reached from 5.5 mg/L in run 1 to 7.1 mg/L in run 3) and increased the levels of SVI by 13.2 mL/g in the aeration zones. As per the study of microbial dynamics, nitrifiers were prevailing at 20 days SRT. The authors of [40] observed better TN removal at high

SRTs; however, [23] confirmed that the decrease in SRT from 37.2 to 27.8, 19.0, and 10.0 days had no characteristic effect on SND efficiencies. In such a way, it can be summarized that reducing SRT for up to 10 days in any treatment system in Indian locations not only improves the biological nutrient removal performance but also benefits the plant from different perspectives.

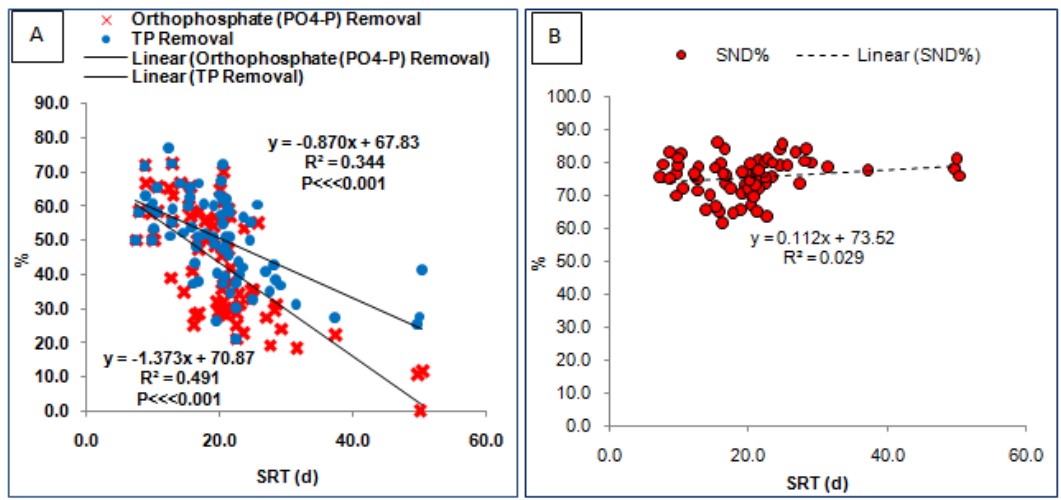

**Figure 5.** Relationships showing (**A**) PO$_4$-P removal and TP removal versus SRT, and (**B**) SND% versus SRT.

**Table 4.** Summary of statistical analyses.

| S. No. | Regression Equations with Std Err. For Coefficient | Factor x | Factor y | F Value | $p$ Value of F Test | $R^2$ | Adjusted $R^2$ | $p$ Value of $t$ Test for Intercept | $p$ Value of $t$ Test for x |
|---|---|---|---|---|---|---|---|---|---|
| 1 | $y = -1.373 \times x + 70.874$ (0.166, 3.622) | SRT | PO$_4$-P Removal% (SRP Removal%) | F (1,72) = 68.647 | $4.95 \times 10^{-12}$ | 0.491 | 0.484 | $2.59 \times 10^{-30}$ | $4.95 \times 10^{-12}$ |
| 2 | $y = -0.871 \times x + 67.831$ (0.143, 3.118) | SRT | TP Removal% | F (1,72) = 37.229 | $5 \times 10^{-8}$ | 0.344 | 0.335 | $4 \times 10^{-33}$ | $5 \times 10^{-8}$ |
| 3 | $y = -0.947 \times x + 77.130$ (0.148, 3.232) | SRT | SOP Removal% | F (1,72) = 41.060 | $1.4 \times 10^{-8}$ | 0.366 | 0.357 | $1.22 \times 10^{-35}$ | $1.4 \times 10^{-8}$ |
| 4 | $y = -0.816 \times x + 90.580$ (0.199, 4.345) | SRT | SAHP Removal% | F (1,72) = 16.900 | 0.00011 | 0.192 | 0.180 | $5.51 \times 10^{-32}$ | 0.00011 |
| 5 | $y = -0.231 \times x + 64.701$ (0.334, 7.093) | SRT | PRP Removal% | F (1,72) = 0.534 | 0.467 | 0.007 | −0.0065 | $5.06 \times 10^{-14}$ | 0.467 |
| 6 | $y = -0.746 \times x + 98.140$ (0.152, 3.314) | SRT | PAHP Removal% | F (1,72) = 24.230 | $5.4 \times 10^{-6}$ | 0.254 | 0.244 | $1.08 \times 10^{-41}$ | $5.4 \times 10^{-6}$ |
| 7 | $y = -1.109 \times x + 63.773$ (0.171, 2.733) | SRT | POP Removal% | F (1,72) = 42.175 | $9.8 \times 10^{-9}$ | 0.372 | 0.364 | $7.35 \times 10^{-27}$ | $9.8 \times 10^{-9}$ |

*3.4. Effect of C: P on TP Removal at Low SRT*

The C/P ratio is crucial for TP and orthophosphate removal in treatment plants. The COD/P ratio should be >45 as per the recommendations in the literature, and the wastewater had an average value of 79.8 and 37.2 BOD/P. Readily active carbon, i.e., the rbCOD fraction was found to be critical among all the COD fractions in affecting BPR. It was identified that the rbCOD/P should be 10–20 for PAO dominance. Higher rbCOD/P (>50) motivates GAOs dominance introducing no or lesser EBPR; moreover, lower rbCOD/P was found insufficient for proper VFA formation in the anaerobic zones and PHB development in the selector zones, which deteriorates the BPR mechanism for taking up sufficient orthophosphates [2,24]. The rbCOD/P in the plant was observed to be 9.3 ± 2.6 (P/C (as rbCOD) ~0.11) (Figure S4, Supplementary Materials). Higher BPR efficiency was gained at low, i.e., 10–12-day SRTs (TP in effluent <~1.5 mg/L), although the plant could have achieved TP in effluent <1.0 mg/L at lower SRTs with higher rbCOD/P ratios (10–20). In

several treatment plants, reduced SRTs and optimum C/P ratios were found to be significant parameters for achieving great BPR. The authors of [41] observed 76.1% TP removal at a C/P ratio of 42.8. The authors of [42] observed that PAOs dominated at low acetate concentrations and higher pH (7.4–8.4). As per [42], GAOs utilize a similar metabolism to PAOs, but they do not accumulate polyphosphate. Circumstances thought to favor GAOs include low pH values (less than approximately 7.25), warmer temperatures (greater than approximately 25 °C), P limitation, glucose addition, and longer solid residence times (SRTs).

### 3.5. Phosphate Release and Uptake at Different SRTs and Runs

The phosphate release and uptake rates were observed by lowering the SRT (Figure 6). Although the rbCOD/P was not high (<10), it favored good BPR efficiencies (~65%).

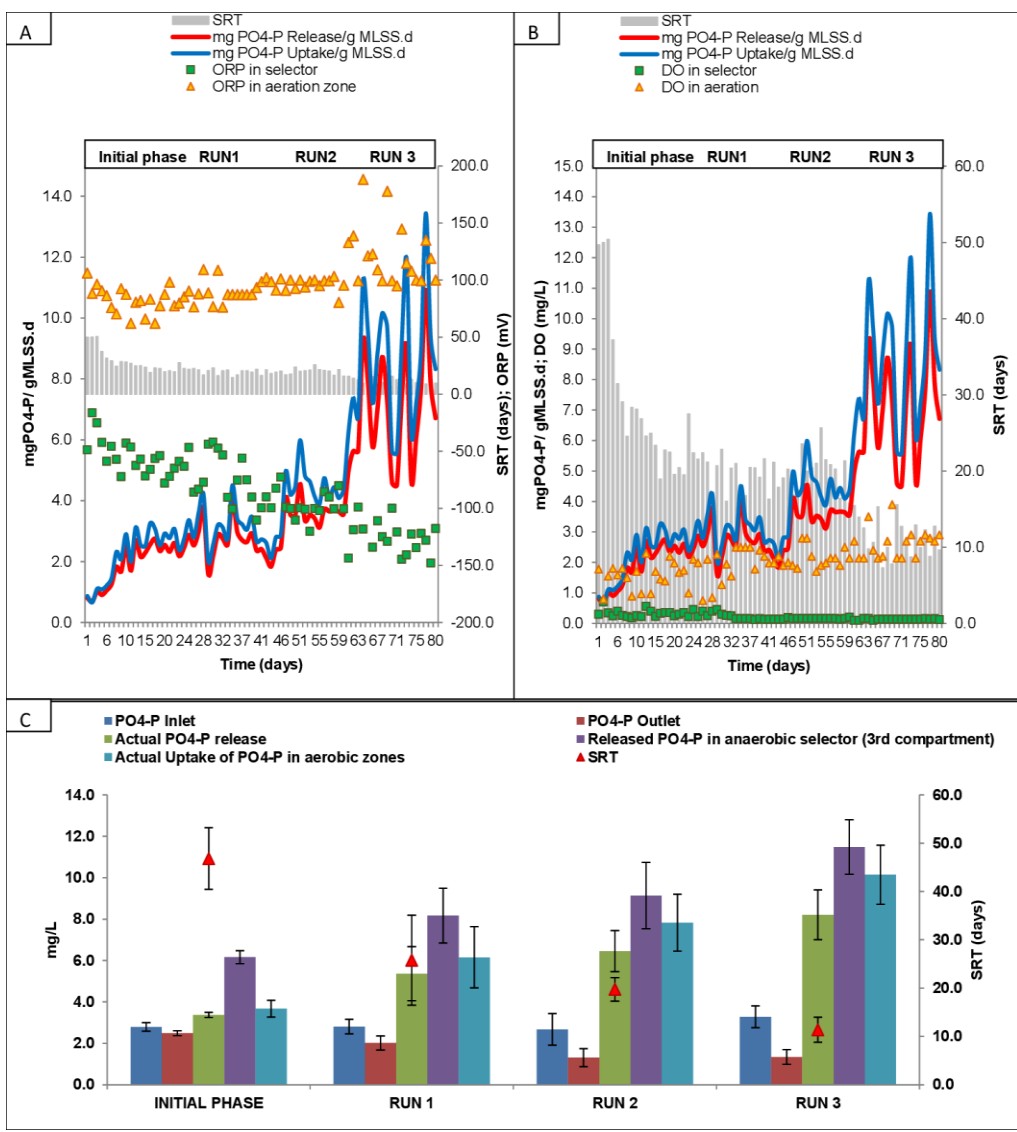

**Figure 6.** PO$_4$-P release and uptake rates at various SRTs along with (**A**) ORP (**B**) DO (**C**) concentration profiles in different phases or zones of SBR.

The orthophosphate release rates in the selector zones reached the maximum of 6.9 ± 1.8 mgPO$_4$-P/gMLSS.d and uptake rates achieved an optimum up to 8.6 ± 2.2 mgPO$_4$-P/gMLSS.d at ~11.4 days SRT (Run 3) in the SBR aeration tanks (13.4 mgPO$_4$-P/gMLSS.d maximum uptake was observed at 8.8 d SRT). Figure 6 sym-

bolizes that on decreasing SRT the orthophosphates uptake rates grew up, but the rate of increase was low; the reason behind this might be the low VFA formation in the wastewater (short lengths of sewer lines). Lesser PAOs developed and flourished in the plant at a quite low available C/P environment even during low SRTs. The ORP in the anaerobic selector zones reached −150 mV, which also depicts the lack of acetates or carbon sources that formed PHBs at a low pace in the anoxic cum anaerobic selector zones (Figure S5, Supplementary Materials). Moreover, DO in the anoxic–anaerobic selector zones was <0.14 mg/L at around 10–15 days SRT and reached ≥2.5 mg/L in the aerobic phase. The authors of [43] worked on full-scale EBPR plants where they observed a 2.2–19.0 mg/gVSS/h aerobic P-uptake rate in various EBPR plants in the Netherlands. In another study by [44], the total P uptake was <8 mg P/g MLSS while using adapted sludge and total P uptake in adapted long-term sludge was <6 mg P/g MLSS (at differently dosed nitrate conditions) which was similar to the present study (at variable SRTs). It was observed in the last compartment of selector zones that actual orthophosphates release (excluding influent $PO_4$-P) in the plant were: 3.4 ± 0.1 mg/L (46.8 ± 6.4 d SRT; initial phase), 5.4 ± 1.3 mg/L (25.8 ± 9.3 d SRT; run 1), 6.5 ± 1.0 mg/L (19.7 ± 2.4 d SRT; run 2), and 8.2 ± 1.2 mg/L (11.4 ± 2.6 d SRT; run 3); while in the SBR aerobic zones the actual uptake was 3.7 ± 0.4 mg/L, 6.2 ± 1.5 mg/L, 7.8 ± 1.4 mg/L, and 10.1 ± 1.4 mg/L, respectively.

### 3.6. TP Fractions and Their Removal in Different Runs with Variable SRTs

During biological treatment, there are various fractions of total phosphorus in wastewater that contribute a major portion of the final treated effluent, and concurrently, the anaerobic/anoxic–aerobic processes and tertiary treatment systems have a significant role in removing the reactive, organic, and particulate acid- hydrolysable fractions of total phosphorus specifically [33,45,46]. The SBR plant at 3 MLD-Roorkee, Uttarakhand, was found to have a lesser removal efficiency of 48 ± 23% due to the continuous nitrate mixing in the strict anaerobic zones and insufficient rbCOD/TP ratios in the influent. However, the disc filtration process after biological treatment in 3 MLD SBR removed the soluble organic phosphorus (SOP) and particulate acid-hydrolysable phosphorus (PAHP) at higher rates than the other fractions due to their adsorption phenomena. In 3 MLD STP, during high SRT of >50 days, the maximum removal was observed for PAHP >80%, and comparatively overall low removal efficiencies were observed in soluble reactive phosphorus (SRP), i.e., 25–35% and soluble and particulate organic phosphorus (SOP and POP) fractions (29–49%). The removal of an average of 68.7% and 55.8% were noticed in the soluble acid-hydrolysable phosphorus (SAHP) and particulate reactive phosphorus (PRP) concentrations in the treatment plants due to their obstinate/refractory characteristics. Lowering the SRT to 10 days had some impacts on TP fractions removal which can be seen in Figure 7. The removal of SRP from 30.2 ± 7.7% in run 1 to 50.6 ± 10.8% in run 2 to 59.2 ± 8.6% in run 3 was observed. The highest removal efficiencies after SRP removal was observed by decreasing SRT from 20 to 15 days in the case of SOP removal (~15.9% increment by switching from run1 to run 2), and POP (~20.5% increment in removal efficiencies by switching from run 1 to run 2).

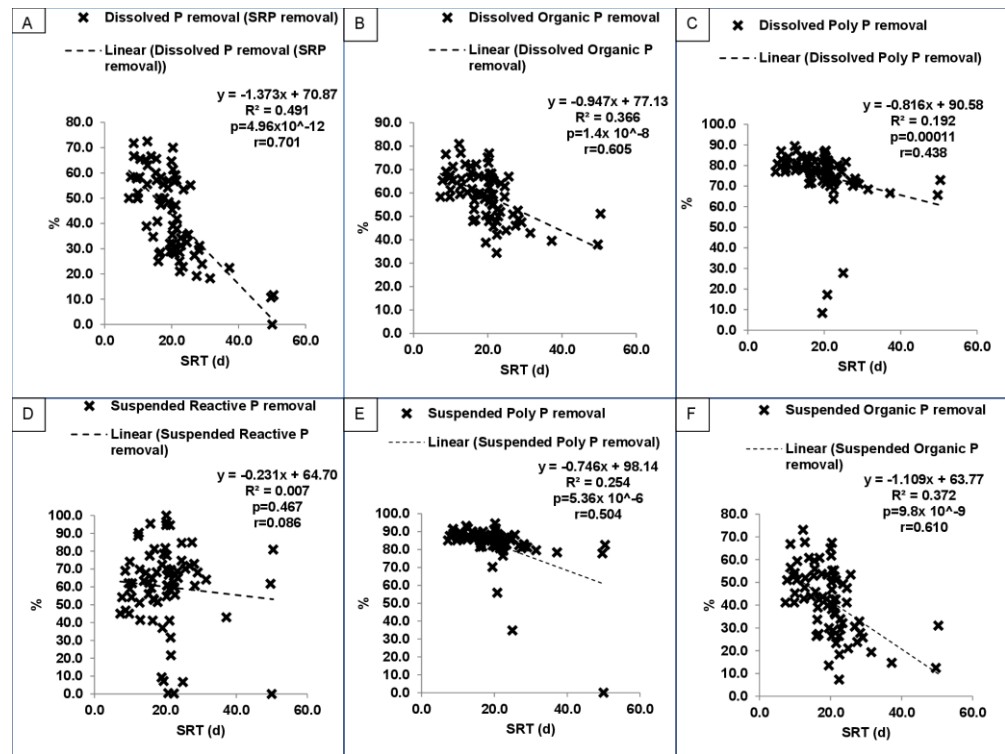

**Figure 7.** Relationships between TP fraction removal%, i.e., (**A**) SRP Removal%, (**B**) SOP Removal%, (**C**) SAHP or Dissolved Poly P Removal%, (**D**) PRP Removal%, (**E**) PAHP or Suspended Poly P Removal%, (**F**) POP Removal% and SRT. Where, Dissolved represents soluble and suspended represents particulate, poly-P is acid-hydrolysable phosphorus or polyphosphates, and SRP is soluble reactive phosphorus or orthophosphates in filtered sample.

It was also observed that further decreasing the SRT from Run 2 to Run 3 did not much improve the system's removal efficiency rates at low rbCOD/TP influent characteristics, as in the case of PRP (~9.8% increment was observed from run 1 to run 2; however, ~3.9% reduction was observed from run 2 to run 3). Additionally, the results observed in SAHP (from ~9.3% increment from run 1 to run 2, and only ~4.2% increment further from run 2 to run 3) were similar to those in PAHP (~2.2% increment from run 2 to run 3), SOP (~2.7% increment from run 2 to run 3), POP (~4.8% increment from run 2 to run 3), and SRP (~8.6% increment from run 2 to run 3). It can be concluded that at low rbCOD/TP ratios, even reducing SRT to <10–12 days do not greatly affect the removal of other TP fractions except SRP. Acid hydrolysis and digestion, which transforms the compressed and condensed phosphates and organic phosphates, i.e., PRP, POP, or PAHP, into orthophosphates, does not classically occur in a biological process such as activated sludge or SBR. Therefore, the fraction of the phosphorus soluble non-reactive phosphorus (SNRP), i.e., SOP and SAHP, cannot be easily eradicated with wastewater-treatment processes. Moreover, incoming VFAs are equally important for high EBPR efficiencies and triggering the EBPR mechanism through PAOs.

According to the observation, there is a need to probe deeper into the bio-selector-attached SBR technologies for removing POP, and PRP fractions, in particular, among all the bio-available forms of P, along with SRT, to satisfy the effluent discharge levels concerning the proper rbCOD fraction of COD to TP ratios, nitrate recirculation in the anaerobic selectors, oxidation-reduction potential in the anaerobic/or anoxic–oxic phases, and the hydraulic retention times of the strict anaerobic zones. There was an overall decreasing dependency observed for the TP fraction removal with SRT. A statistically significant relationship ($R^2$ ~0.5 and $p < 0.001$) was observed for the SRP fraction of TP and SRT as depicted in Table 4.

### 3.7. Light Microscopy and SEM Analysis for Microbiota Identification in Biomass

There are different microbes present in the aeration sludge of SBR tanks governing organic matter and nutrient removal. It contains filamentous species such as *Micothrixparvicella*, *Thiothrix*, *Bacillus*, *Nocardia*, *Pseudomonas*, and *Rhizobium* which accumulate PHBs in their cells [47]. Several species are also present which can be observed in SEM analysis such as *coccoid*, *bacilli*, and *spirillum* in the biomass (Figure 8). Additionally, protozoa including Arcella, Vorticella, and Rotaria were observed to be dominant through light microscopy, indicating excellent settling properties of sludge with lesser SVIs (<80 mL/g) as depicted in Table S1 (Supplementary Materials) [25]. *Quadricoccus australiensis*, generally coccus or cocci found in tetrads, resemble the group of bio P-bacteria commonly seen in activated sludge and stained positively for intracellular polyphosphates and PHB. *Defluviccocus* (11.6%) and *Tessaracoccus* (0.1%) are facultative anaerobic genera found as cocci in tetrads and responsible for aerobic P removal, and denitrification and phosphorus accumulation, respectively, in plants at lower SRTs [48]. *Ralstonia eutropha* belonging to the *Burkholderia* family (7.7%) are also observed capable to remove phosphorus and R. Eutrophus produces PHAs in activated sludge plants [49] (Figure 9B). Figure 9B depicts *Burkholderia* in the plant.

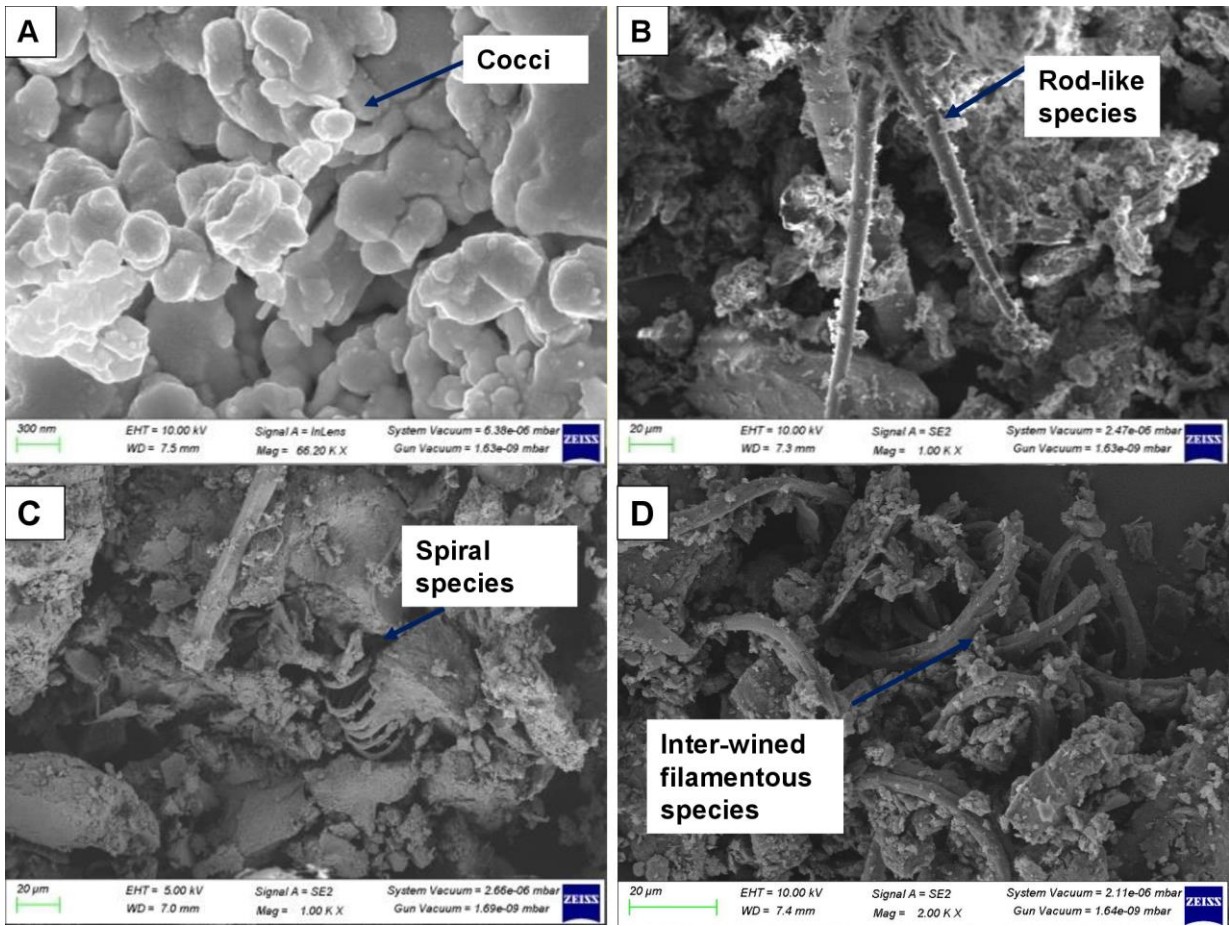

**Figure 8.** SEM images of dry activated 3 MLD SBR sludge representing (**A**) Cocci structures (**B**) Rod-like structures (**C**) Spiral species and (**D**) Inter-wined filamentous species.

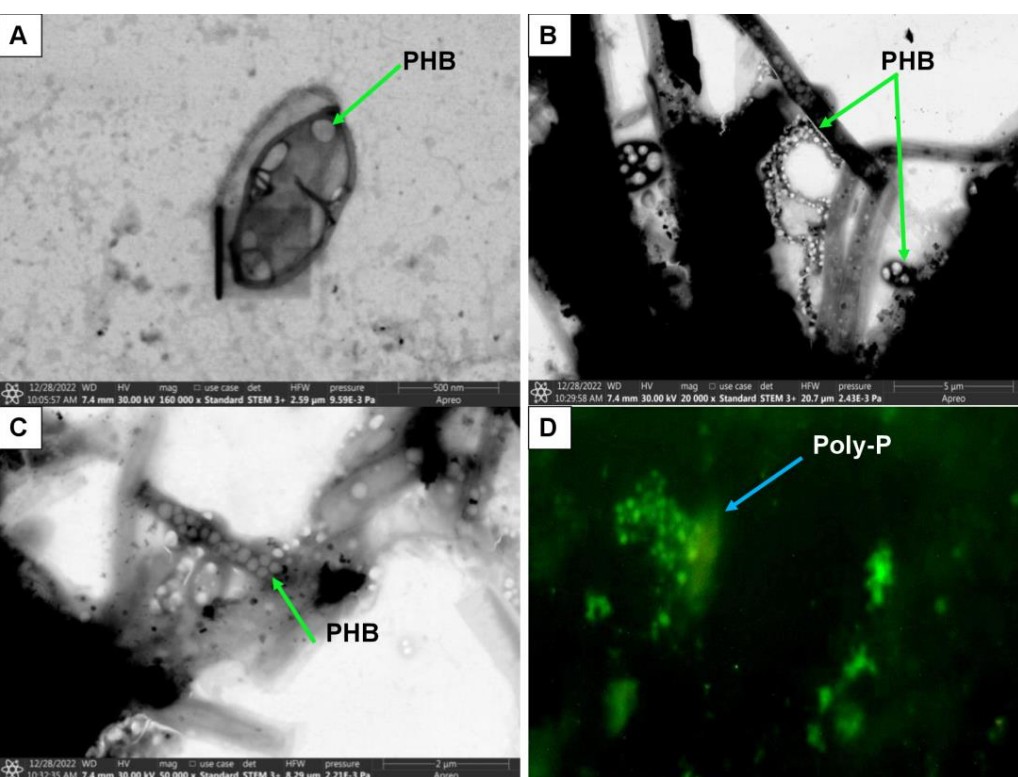

**Figure 9.** (**A–C**) intracellular PHB granules inside the flocs of sludge, (**D**) DAPI-stained polyphosphate globules (bright-green and yellow) in the biomass of 3 MLD SBR at optimum SRT during run 3.

### 3.8. Microscopic Investigations for Intracellular Polymers Development: Light Microscopy and STEM Analysis

PHBs and polyphosphates are intracellular storage products, formed during the EBPR mechanism. It has been found that denitrifying bacteria and PAOs prefer readily biodegradable organic material such as VFAs in N and P removal [5,50]. The sequencing batch reactor (SBR) systems are effective in producing and accumulating PHBs and governing SND and EBPR processes. The PHBs are electron donors both for EBPR and SND processes. The feast and famine strategy in a SBR mode was the main approach used for this purpose and had shown to benefit biomass growth and PHB accumulation. In the feast phase, the VFAs are stored as PHBs, exogenous BOD is consumed, and orthophosphates are liberated from the cell to the wastewater under anaerobic conditions, while in the famine phase, these stored PHBs are degraded to be used as a carbon source for accomplishing metabolic activities of bacteria and consecutively excess polyphosphates (as EBPR) are taken up in the aerobic zones [51]. Similarly, [52] reported that an increase in PHA production in activated sludge leads to enhancing denitrification efficiency. Thus, the availability of internal electron donors as storage compounds (PHBs) significantly affects N removal by SND in SBR [30,52].

### 3.9. 16SrRNA Sequencing and Microbial Community: Before and after Modifications

The putative PAO species which are observed for phosphorus accumulation are shown in Figure 10. It is easy to comprehend that the abundance of these species was dominant during operations at a lesser SRT for 15–20 days than at greater SRTs ranging >40–50 days. *Candidatus Accumulibacter* and *Tetrasphaera* are the most commonly found PAOs in any treatment system performing excellent phosphorus removal; however, their abundance was low in the plant while denitrifying phosphorus-removing organisms were dominant in the out analyses of the aerobic biomass. *Defluvicoccus*, *Rhizobiales*, and *Burkholderia* were observed as dominant genera contributing to P removal in the plant [48]. *Nitrospira* and *Nitrosomonas* were dominant among all at 20 days' SRT, justifying excellent nitrification

rates and high SND (75–80%) [53]. *Microlunatus phosphovorus* (0.07%, 0.02%), *Pseudomonas* (0.07%, 0.12%), and *Paracoccus denitrificans* (1.9%, 2.3%) were found in the plant's sludge at high and low SRTs, respectively. *Dechloromonas* can eliminate the phosphorus via the denitrifying phosphorus-removal pathways and achieve simultaneous nitrogen and phosphorus removal [10]. The literature suggests that an SRT of 15 days is optimum for *Dechloromonas* as they have slower growth rates [54]. An SRT of 18–20 days was observed to be conducive to DPAO proliferation in an A2NSBR system [9]. Ottowia, Bosea, and Shinella have also performed enhanced concurrent biological N and P removal in plants [55]. Microbial communities responsible for EBPR systems are contributing to conventional PAOs, supplemented by GAOs, and the major groups such as Rhizobiales (14.0% and 11.0%), and Flavobacterium (2.7% and 2.2%), during low and high SRTs, respectively, in the SBR. Additionally, nitrifiers such as *Nitrospira* and *Nitrosomonas* were abundant, denitrifiers such as *Paracoccus*, and *Azoarcus* were leading, and others were lignin degraders and acid-producing and nitrogen-fixation-causing organisms (Figures S6 and S7 (Supplementary Materials)).

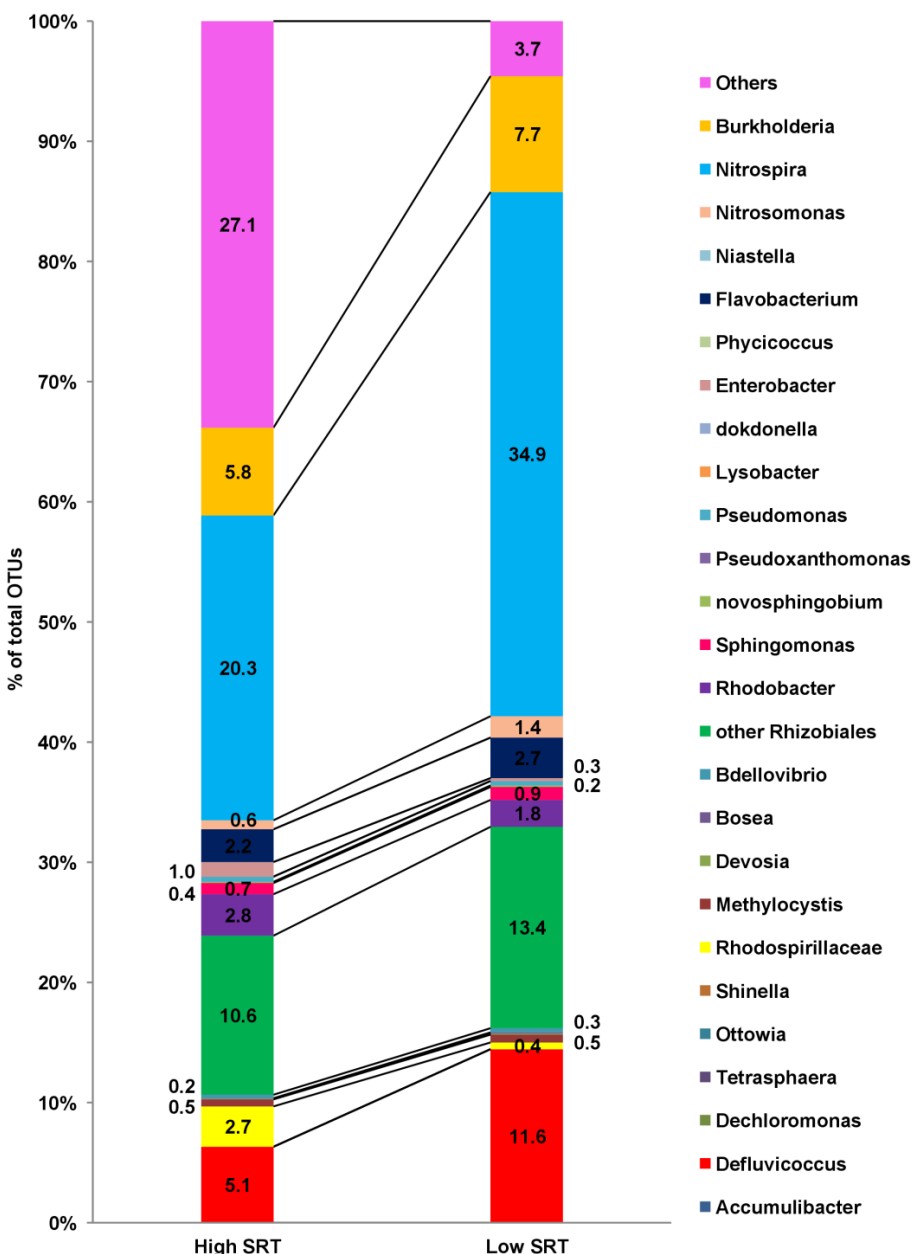

**Figure 10.** Microbial population dynamics at two different SRTs (comprising specifically 'nitrifiers' and 'putative PAOs'). High SRT (≥50 days) and low SRT (~20 days).

### 3.10. Dewatered Sludge Parameters: Before and after Modifications in SRT

Dewatered sludge parameters were observed to check the quality of sewage sludge produced after dewatering at variable SRTs. It was observed that reducing SRT had some positive increasing effects on calorific value, total organic carbon content, N-P-K, and C/N ratios while reducing the pathogenic activities in terms of fecal coliforms, salmonella, and helminth eggs. The moisture content, total organic carbon (TOC), and the sludge's phosphorus/phosphate content were found to be slightly increased by 10.3%, 27.5%, and 42.7%, respectively, in the SBR plant at low SRTs, whereas pathogens were decreased, including fecal coliforms by 16.3%, salmonella by 25.0%, and helminth eggs by 14.1%, by lowering the SRT from 50 to 15 days. The sludge nutrient quality and calorific value parameters were observed to be optimum according to FCO standards; however, the pathogen level in the plant exceeds the criteria of USEPA and CPHEEO standards (Table 5). Eventually, it can be estimated by the records that lowering the SRTs enhances the performance of the whole system.

**Table 5.** Dewatered sludge characteristics at different SRTs.

| Parameters | FCO Composts Standards (2009) | US EPA Class a Sludge | USEPA Class b Sludge | CPHEEO-2013 | 3 MLD (at High SRT 50 Days) Dewatered Sludge | 3 MLD (at a Low SRT 10 Days) Dewatered Sludge |
|---|---|---|---|---|---|---|
| pH value | 6.5–7.5 | - | - | - | $6.8 \pm 0.3$ | $7.3 \pm 0.1$ |
| Moisture Content (%) | 15–25 | - | - | - | $75.0 \pm 1.5$ | $82.7 \pm 1.3$ |
| Color | Dark brown to black | - | - | - | Dark brown to black | Dark brown to black |
| Odor | Absence of foul odor | - | - | - | Absence of foul odor | Absence of foul odor |
| Bulk Density (g/cm$^3$) | <1 | | | - | $1.3 \pm 0.2$ | $1.3 \pm 0.1$ |
| Conductivity (ds/m) | Not more than 4 | - | - | - | $0.9 \pm 0.2$ | $0.8 \pm 0.3$ |
| Total Organic Carbon (%) | $\geq 12$ | - | - | - | $24.7 \pm 1.5$ | $31.5 \pm 1.3$ |
| Total Nitrogen (%) | $\geq 0.8$ | - | - | - | 1.6–3.0 | 1.9–2.4 |
| Total Phosphate as P$_2$O$_5$ (%) | $\geq 0.4$ | - | - | - | 2.2–2.9 | 3.7–5.2 |
| Total Potassium as K$_2$O (%) | $\geq 0.4$ | - | - | - | $0.3 \pm 0.03$ | $0.4 \pm 0.05$ |
| C/N ratio | <20 | - | - | - | $9.7 \pm 1.2$ | $14.6 \pm 0.6$ |
| Particle Size | 90% materials 4 mm sieve | - | - | - | - | - |
| Fecal Coliforms (MPN/g dry solids) | - | <1000 MPN/g | $<2 \times 10^6$ MPN/g | $<2 \times 10^6$ MPN/g | $43,000 \pm 570$ | $36,000 \pm 430$ |
| Salmonella species (MPN/4 g dry solids) | - | <3 MPN/4 g | - | - | $8.0 \pm 1.0$ | $6.0 \pm 1.0$ |
| Helminth eggs (Numbers/4 g of TS) | - | ≤1 per 4 g of total solids (dry weight basis) | - | - | $64 \pm 5$ | $55 \pm 2$ |
| Gross Calorific Value (Kcal/kg) Dry Basis | - | - | - | - | $2250 \pm 280$ | $2281 \pm 190$ |
| SOUR (mgO$_2$/g TS.h) Vector Attraction | - | <1.5 | - | - | $1.3 \pm 0.3$ | $1.2 \pm 0.1$ |

### 3.11. Reduction in Energy-Consumption Rate at Low SRT

The utilization of a considerable amount of energy in the maneuver of WWTPs would worsen conditions, as wastewater-treatment plants (WWTPs) consume about 1–3% of the entire electric energy production [56]. By adopting modern technologies, certain process modifications, and optimization techniques, energy efficiency can be improved, consuming less energy for WWTP operation [57,58].

It has been stated that significantly high SRTs increase oxygen requirements and energy usage in treatment plants [36]. SRTs that are excessively low (<8 days) deteriorate the

nitrification and denitrification efficiency, and those that are too high (>30 days) cause poor EBPR performance as the phosphorus uptake rates are reduced [36,59]. SRTs of 8–15 days were found to be adequate for biological nutrient removal and excellent sludge settling characteristics. The classic power consumption of wastewater/sewage treatment for one MLD of wastewater using ASP ranges from 1000 to 2500 mega joule (MJ, 1000 to 25003106 J, or 277.8 to 694.4 kWh) [31,58]. In this study of optimizing SRT with energy consumption, the rates were observed as 806.8 ± 44.4 KWh/d (2.3 ± 0.18 MLD, run 1), 781.3 ± 79.4 KWh/d (2.40 ± 0.15 MLD, run 2), and 745.0 ± 68.5 KWh/d (2.40 ± 0.15 MLD, run 3). During this study, four parameters were varied: (a) overall SRT of the system (from 56 days to <10 days), (b) DO levels in the selector zones reduced (as diffusers' running time was reduced from 15 min to 10 min per cycle), (c) SAS flow rates (SAS capacity, i.e., 35 m$^3$/h*SAS running time (h)/d), and (d) operating MLSS of the aeration sludge. Figure 11 shows the run-wise results and Figure S8 (Supplementary Materials) gives the relationship between energy consumption and operational parameters. The sludge production (dry solids produced) in the plant was 157.3 ± 20.4 kg/d (initially), 221.0 ± 33.8 kg/d (run 1), 227.6 ± 36.4 kg/d (run 2), and 242.2 ± 25.3 kg/d (run 3). As SRT was reduced to 10 days, overall energy consumption was lowered, there was a reduction in aeration sludge MLSS, and DO in the selector zones was affected in a similar way to the overall system. Increasing SAS flow rates benefitted the system by reducing the energy-consumption rates. The authors of [58] recommend that optimizing SRT from 8 to 12 days promotes the energy efficiency of wastewater-treatment systems.

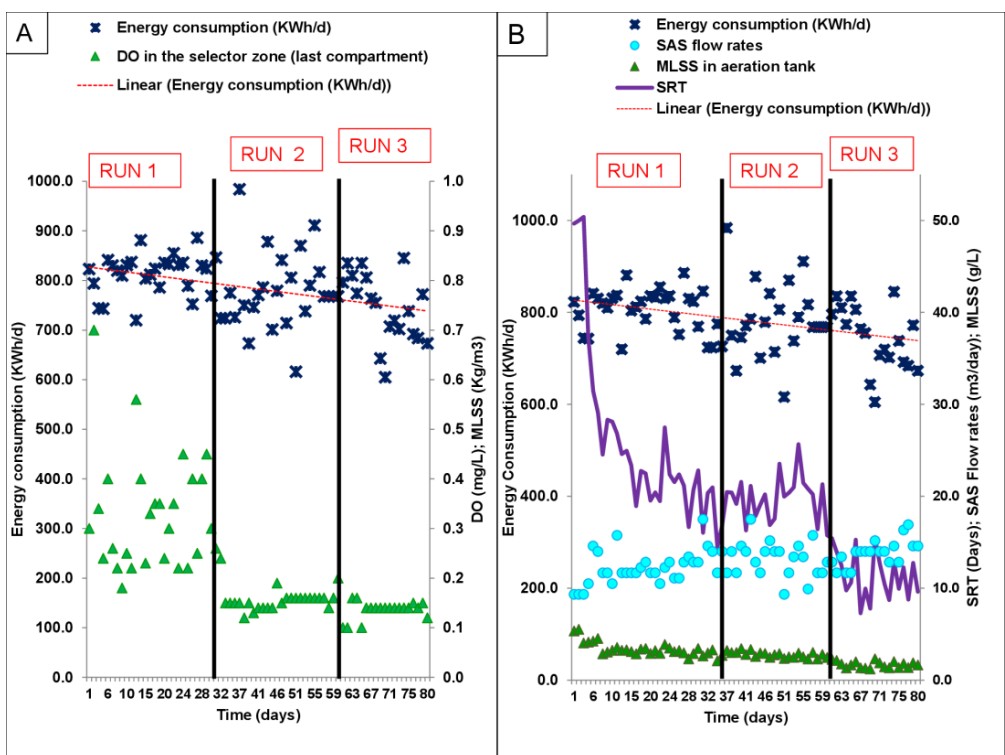

**Figure 11.** (**A**) Energy consumption/day versus DO, and (**B**) Energy consumption/day versus SRT, MLSS and surplus (waste) activated sludge (SAS) flow rates.

## 4. Conclusions

The SRT has played a significant role in selector-attached SBR systems with low-influent readily biodegradable carbon-source-to-phosphorus (C/P) ratios. Five critical results were observed in the study: (a) Simultaneous nitrogen and phosphorus removal was observed in the plant by optimizing it at 11.4 ± 2.6 days SRT; SND% was 76.5 ± 5.5% and TP removal efficiency was observed 61.2 ± 7.6% at a DO control of 0.5–2.5 mg/L in the SBR aeration tanks. (b) PHB and polyphosphates are intracellular polymers produced during SND and

EBPR processes. (c) *Nitrospira* and *Nitrosomonas* were dominant nitrifiers; and *Defluvicoccus*, *Rhizobiales*, and *Burkholderia* genera were the dominant DPAOs observed at 20 days SRT in the SBR. (d) Dewatered sludge characteristics suggest that C/N ratio, phosphorus content, and pathogen-removal efficiencies improve as the SRTs were reduced to ten days. Last but not least, (e) energy-consumption rates were also minimized by lowering the SRTs to ~11.4 days, on an average, in run 3. In a nutshell, the study implies that not only nutrient removal but the overall system's performance can be improved by optimizing it at $11.4 \pm 2.6$ days SRT, at a low rbCOD/TP of ~9.3 < 10 and strict anaerobic conditions in pre-selector zones (DO < 0.15 mg/L) in cyclic-technology-based SBRs. This can be a remarkable and promising approach to optimize existing full-scale treatment plants and design novel pre-anoxic selector-attached SBR plants by SRT and DO control, specifically ensuring EBPR complies with nitrogen removal, for developing countries such as India.

**Supplementary Materials:** The following supporting information can be downloaded at: https://www.mdpi.com/article/10.3390/su15107918/s1, Figure S1: The ORP, DO and pH profiles with SRT in subsequent runs of the SBR plant (aeration tank and selector's last compartment); Figure S2: Wastewater fractions in terms of COD, BOD, TSS, and TP during the SRT > 50 d as described in [25]; Figure S3: SRT calculated by two ways and average minimum SRT observed; Figure S4: Profiles of $PO_4$-P and TP removal with SRT and (a) rbCOD/TP (b) COD/TP; Figure S5: Cycle-wise profile of DO, ORP and $PO_4$-P in 3 MLD SBR in different runs; Figure S6: (A) Phylogenetic tree analyses of top 30 species in Aeration Sludge of 3 MLD Full-scale SBR at 20 days SRT, and (B) Functional micro-organisms prevalence in aeration SBR sludge at 20 days SRT; Figure S7: Venn diagram showing relationship between species observed in Aeration Sludge of 3 MLD Full-scale SBR at >50 days SRT and 20 days SRT; Figure S8: Relationships among energy consumption and SRT, aeration MLSS, WAS/ SAS flow rates and DO in selector zones; Table S1: Protozoa identification in 3 MLD SBR.

**Author Contributions:** Conceptualization, G.S.; methodology, G.S.; software, G.S.; validation, G.S., A.K., and A.A.K.; formal analysis, G.S.; investigation, G.S.; resources, G.S., A.K. and A.A.K.; data curation, G.S.; writing—original draft preparation, G.S.; writing—review and editing, G.S. and A.A.K.; visualization, G.S. and A.A.K.; supervision, A.A.K.; project administration, A.A.K.; funding acquisition, G.S. All authors have read and agreed to the published version of the manuscript.

**Funding:** The first and corresponding author is getting funds from Ministry of Human Resource Development (MHRD) for Ph.D. Assistantship at IIT Roorkee. The first author is also thankful to Department of Science and Technology, GoI (Grant No. DST/IMRCD/India-EU/Water Call 2/SARASWATI 2.0/2018/C) for financial support to the research.

**Institutional Review Board Statement:** Not applicable.

**Informed Consent Statement:** Not applicable.

**Data Availability Statement:** All the data is available in manuscript and Supplementary Information.

**Conflicts of Interest:** The authors declare no conflict of interest.

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
