# Peer review of "Improved Biological Phosphorus Removal under Low Solid Retention Time Regime in Full-Scale Sequencing Batch Reactor"

_sustainability, doi:10.3390/su15107918_

Round 1
Reviewer 1 Report
This manuscript presents the improvement of the P-removal conditions biologically from a full-scale SBR plant, simultaneously accomplishing high COD, BOD, TSS, and nitrogen removal (via SND). The subject of this paper is well within the scope of the journal. In my opinion, it can further proceed after small improvements such as minor revision.
At section 3.5. Phosphate release and uptake at different SRTs and runs
- in Figure A the legend is incomplete (cut off the last line)
- Figure 7. Relationships between TP Fractions removal% and SRT C and D should be optimized so as not to overlap the equations over the points
- Figure 11. (a) Energy consumption/ day in different SRTs; (b) Energy consumption/ day versus DO, 575 and MLSS, and (c) Energy consumption/ day versus surplus (waste) activated sludge (SAS) flow 576 rates. - please otimized the Figure 11 (b) the legend is incomplete (cut off the last line)
Author Response
Dear Editor and Reviewers,
Thank you for your valuable comments/ feedback in our study. It is our great privilege to gratify you replying your comments. Please, have a look to our answers. We hope that we have satisfied all your questions/ queries.
|
1) Reviewer 1:
|
||
|
Q. No. |
Comments |
Authors’ answers to Reviewers’ comments |
|
1. |
This manuscript presents the improvement of the P-removal conditions biologically from a full-scale SBR plant, simultaneously accomplishing high COD, BOD, TSS, and nitrogen removal (via SND). The subject of this paper is well within the scope of the journal. In my opinion, it can further proceed after small improvements such as minor revision. |
Thank you for your valuable comments and suggestions. |
|
2. |
At section 3.5. Phosphate release and uptake at different SRTs and runs- in Figure A the legend is incomplete (cut off the last line) |
Corrected |
|
3. |
Figure 7. Relationships between TP Fractions removal% and SRT C and D should be optimized so as not to overlap the equations over the points |
Corrected |
|
4. |
- Figure 11. (a) Energy consumption/ day in different SRTs; (b) Energy consumption/ day versus DO, 575 and MLSS, and (c) Energy consumption/ day versus surplus (waste) activated sludge (SAS) flow 576 rates. - please optimized the Figure 11 (b) the legend is incomplete (cut off the last line). |
Corrected and completed |

Reviewer 2 Report
The article is interesting and has practical implications, but I would suggest that it be cleaned up a bit before publication. My main points:
1. The title should be shortened. The title must not contain abbreviations.
2. There are too many keywords written too broadly.
3. Review the citation requirements for publications. Publications must be numbered in square brackets.
4. Line 62: does not explain what the PHB is.
5. Line 66: does not explain what SVI is.
6. Line 79: not phyla, but phylum.
7. Line 126: does not explain what NGT is.
8. Line 127: does not explain what CPHEEO is.
9. There is too much self-citation: the reference list contains 5 Srivastava G. et al. publications.
10. Figure 2: The title does not explain what is shown in parts A, B and C.
11. Figure 2 and Figure 3 are barely discussed, the results should be commented more deeply.
12. Figure 4: The title does not explain what is shown in parts A, B, C and D.
13. Figure 5: Messy y-axis name.
14. Table 4 is uncommented.
15. Lines 501-513: The font size changes.
16. The authors applied 11.4 ± 2.6 days SRT and obtained a TP removal efficiency of about 61%. However, with a much longer SRT, more efficient phosphorus removal was obtained by authors (MažeikienÄ—, A., & Grubliauskas, R. 2021. Biotechnological wastewater treatment in small-scale wastewater treatment plants. Journal of Cleaner Production, 279, 123750.): ,,Excess sludge was removed from the small plant only twice in ten months, but effectiveness in elimination of phosphorus 82% was achieved’’. Please comment on such results.
17. The conclusion section should not contain a citation, but rather a cited publication (Srivastava et al., 2021).
Author Response
Dear Editor and Reviewers,
Thank you for your valuable comments/ feedback in our study. It is our great privilege to gratify you replying your comments. Please, have a look to our answers. We hope that we have satisfied all your questions/ queries.
|
1) Reviewer 2 |
||
|
1. |
The title should be shortened. The title must not contain abbreviations. |
Corrected and changed. The new title is: “Improved biological phosphorus removal at low Solids Retention Times regime in full-scale Sequencing Batch Reactor”. |
|
2. |
There are too many keywords written too broadly. |
Corrected. |
|
3. |
Review the citation requirements for publications. Publications must be numbered in square brackets. |
Done. |
|
4. |
Line 62: does not explain what the PHB is. |
Corrected and Added |
|
5. |
Line 66: does not explain what SVI is. |
Added |
|
6. |
Line 79: not phyla, but phylum. |
Corrected |
|
7. |
Line 126: does not explain what NGT is. |
Added |
|
8. |
Line 127: does not explain what CPHEEO is. |
Added |
|
9. |
There is too much self-citation: the reference list contains 5 Srivastava G. et al. publications. |
Corrected and reduced. |
|
10. |
Figure 2: The title does not explain what is shown in parts A, B and C. |
Done |
|
11 |
Figure 2 and Figure 3 are barely discussed, the results should be commented more deeply. |
Added results in the sections 2.2 (i) and (ii). |
|
12 |
Figure 4: The title does not explain what is shown in parts A, B, C and D. |
Added. |
|
13 |
Figure 5: Messy y-axis name. |
Corrected. |
|
14 |
Table 4 is uncommented. |
Comments added. |
|
15 |
Lines 501-513: The font size changes. |
Corrected. |
|
16 |
The authors applied 11.4 ± 2.6 days SRT and obtained a TP removal efficiency of about 61%. However, with a much longer SRT, more efficient phosphorus removal was obtained by authors (MažeikienÄ—, A., & Grubliauskas, R. 2021. Biotechnological wastewater treatment in small-scale wastewater treatment plants. Journal of Cleaner Production, 279, 123750.): ,,Excess sludge was removed from the small plant only twice in ten months, but effectiveness in elimination of phosphorus 82% was achieved’’. Please comment on such results. |
There are several literature and studies, involving, i.e., Lee et al., 2007, Li et al., 2008, Onnis-Hayden et al., 2019, and Wang et al., 2019, cited in the present paper describing that low SRTs have benefitted the EBPR community. Also, at a low rbCOD/ TP ratio (<10), low SRTs have helped the microbial community of PAOs prevail over GAOs. The old and dead sludge (endogenous decay) accumulated at longer SRTs- devoid of/ lack of denitrifying PAOs should be wasted in 3 MLD SBR to gain higher phosphate removal efficiencies as shown in Figures 4, 5, and 10.
As per the study of Mažeikienė, A., & Grubliauskas, R. 2021, three points should be carefully observed:
But, if at long SRTs, dead sludge has accumulated with a lack of PAOs or DPAOs community, then we need to waste it and grow new biomass providing conditions of EBPR (optimum rbCOD/ TP ratios, high negative ORPs in the wastewater coming from long sewer lines, lesser recirculation of nitrates or DO in anoxic-anaerobic selector zones, and SRTs 10-15 days), so that proper growth of PAOs may there. It was the case in the present study, where low SRTs were found capable to achieve high removal efficiencies of Nitrogen and Phosphorus, simultaneously. |
|
17 |
The conclusion section should not contain a citation, but rather a cited publication (Srivastava et al., 2021). |
Corrected. |
Thanking you.

Reviewer 3 Report
Intersting article, important for operational reasons. It requires minor adjustments: line 28: Keywords - please remove "Poly-β-hydroxybutyrate" Line 54: GAOs - there is no expalantion for this term, we can find it in line 82, it is too late. line 81: "polyphosphate-accumulating organ-81 isms (PAOs) " - You have alreday explained it in line 50 Table 2 and 3 - it is difficult to read them line 524 - please remove asterisk table 5 - I am sure that you understand that higher TOC (%) in 10 days sludge means more energy, money for the sludge stabilization? lines 578-598 Conclusions sections - it is written in a way that is not friendly to the reader line 689 - refrence No 44 should be replaced with 44 (because of the year of publication) Final remark - I am curious whether the efficiency of removal of nitrogen, phosphorus and organic substances presented will not change in the long term, when SRT will stay at 10d value?
Author Response
Dear Editor and Reviewers,
Thank you for your valuable comments/ feedback in our study. It is our great privilege to gratify you replying your comments. Please, have a look to our answers. We hope that we have satisfied all your questions/ queries.
|
1) Reviewer 3 |
||
|
1 |
Interesting article, important for operational reasons. |
Thank you for your valuable comments. |
|
2 |
line 28: Keywords - please remove "Poly-β-hydroxybutyrate" |
Removed |
|
3 |
Line 54: GAOs - there is no expalantion for this term, we can find it in line 82, it is too late. |
Added |
|
4 |
line 81: "polyphosphate-accumulating organ-81 isms (PAOs) " - You have alreday explained it in line 50 |
Corrected |
|
5 |
Table 2 and 3 - it is difficult to read them |
Corrected and added new ones. |
|
6 |
line 524 - please remove asterisk |
Removed |
|
7 |
table 5 - I am sure that you understand that higher TOC (%) in 10 days sludge means more energy, money for the sludge stabilization? |
Yes, however, for making compost of good quality, TOC% ≥12 is needed (as per the FCO guidelines, 2009). Higher TOC corresponds to greater fertilizer/ compost value. In such a way, I hope, it can be a positive sign for sludge stabilization if TOC content increases by lowering the SRT to 10 days. |
|
8 |
lines 578-598 Conclusions sections - it is written in a way that is not friendly to the reader |
Correct and improved. |
|
9 |
line 689 - refrence No 44 should be replaced with 44 (because of the year of publication) |
Corrected and the year of publication added. |
|
10. |
Final remark - I am curious whether the efficiency of removal of nitrogen, phosphorus and organic substances presented will not change in the long term, when SRT will stay at 10d value? |
Dear Reviewer, I am continuously monitoring and working on numerous (more than 40) full-scale SBR plants and thoroughly in a pilot-scale plant (during my four years of Ph.D. study). Therefore, I can submit that low SRTs of < 15 days and proper design and operating conditions can achieve excellent nutrient removal efficiencies via SND and EBPR processes in a long run. In my pilot-scale SBR, under project SARASWATI 2.0 (INDIA), 10 days SRT was observed as optimum for achieving effluent TP<1 mg/L and effluent TN< 6 mg/L in Indian conditions. The paper depicting the above pilot-scale operation is written and under the submission phase. Interesting point. As per your suggestions, I am also changing several runs of my pilot plant and introducing higher SRTs to check the performance of the plant. Thanking you. |
Thank you.
Looking forward to your positive response
Best Regards,
Corresponding Author
